# Effect of a multi-faceted rapid response system re-design on repeat calling of the rapid response team

**Richard Chalwin**[1,2]*, **Amy Salter**[1], **Jonathan Karnon**[3], **Victoria Eaton**[2], **Lynne Giles**[1]

**1** School of Public Health, Faculty of Health and Medical Sciences, University of Adelaide, Adelaide, South Australia, Australia, **2** Rapid Response System, Lyell McEwin Hospital, Elizabeth Vale, South Australia, Australia, **3** College of Medicine and Public Health, Flinders University, Bedford Park, South Australia, Australia

* richard.chalwin@adelaide.edu.au

**Data Availability Statement:** All relevant data are within the manuscript and its Supporting Information files.

**Funding:** The author(s) received no specific funding for this work.

## Abstract

### Background

Repeat Rapid Response Team (RRT) calls are associated with increased in-hospital mortality risk and pose an organisation-level resource burden. Use of Non-Technical Skills (NTS) at calls has the potential to reduce potentially preventable repeat calling. NTS are usually improved through training, although this consumes time and financial resources. Re-designing the Rapid Response System (RRS) to promote use of NTS may provide a feasible alternative.

### Methods

A pre-post observational study was undertaken to assess the effect of an RRS re-design that aimed to promote use of NTS during RRT calls. The primary outcome was the proportion of admissions each month subject to repeat RRT calling, and the average number of repeat calls per admission each month was the secondary outcome of interest. Univariate and multivariable interrupted time series analyses compared outcomes between the two study phases.

### Results

The proportion of admissions with repeat calls each month increased across both phases of the study period, but the increase was lower in the post re-design phase (change in regression slope -0.12 (standard error 0.07) post versus pre re-design). The multivariable model predicted a 6% reduction (95% confidence interval -15.1–3.1; P = 0.19) in the proportion of admissions having repeat calls at the end of the post redesign phase study compared to the predicted proportion in the absence of the re-design. The average number of calls per admission was also predicted to decrease in the post re-design phase, with an estimated difference of -0.07 calls per admission (equivalent to one fewer repeat call per 14 patients who had RRT calls) at the end of the post re-design phase (95% confidence interval -0.23–0.08, P = 0.35).

**Competing interests:** The authors have declared
that no competing interests exist.

## Conclusion

This study of an RRS re-design showed modest, but not statistically significant, reductions
in the proportion of admissions with repeat calls and the mean number of repeat calls per
admission. Given the economic and workforce capacity issues that all health care systems
now face, even small improvements in the RRS may have lasting impact across the organi-
sation. For the potential interest of RRS managers, this paper presents a pragmatic, low-
cost initiative intended to enhance communication and cooperation at RRT calls.

## Introduction

Over the past quarter century, the Rapid Response Team (RRT) has evolved from a conceptual
advancement of the response to in-hospital cardiac arrest to become a ubiquitous patient safety
mechanism [1, 2]. Throughout this time, studies, and reviews of Rapid Response System (RRS)
activity have consistently demonstrated increasing rates of RRT calling, as RRS mature and the
hospitals they are based within become busier [2–5].

In some respects, this suggests desirable awareness and utilisation of a patient safety mecha-
nism. Indeed, the increase in RRS usage within an organisation has been associated with
improved patient survival statistics [3, 6, 7]. However, increasing RRS activity poses a logistical
and resourcing burden for hospitals, as most RRTs tend to not be supernumerary, with staff
rostered from other substantive roles [4, 8, 9]. Although adverse effects have not yet been
attributed to team members leaving other duties to attend RRT calls [10], the potential exists
for these to occur. This risk could be magnified during concurrent RRT calls as resources are
typically not available to provide a full response to more than one call simultaneously [9].

Against this background of increasing activity, the RRS should seek efficiencies to facilitate
RRT capacity to promptly attend all unexpected clinical deteriorations. One avenue could be
through reduction of potentially preventable repeat calling, that is the RRT attending a patient
more than once due to inadequate resolution of an initial call, especially when the repeat call
closely follows the first. In a previous study, we found increased mortality risk in patients re-
attended by the RRT within 24 hours of a previous completed call in which clinical issues
remained unresolved [11].

Deficits in non-technical skills (NTS), such as communication and cooperation, at RRT
calls have been identified as a risk factor for potentially preventable repeat calling [11–13].
Effective employment of NTS are crucial due to the inherent time and clinical pressures
imposed by the deteriorating patient [14, 15]. Ideally, NTS would be augmented by delivery of
specialised, simulated scenario training for RRTs [14, 16]. However, such training requires tak-
ing staff away from clinical duties, which is often not feasible in resource-limited hospitals.

Therefore, a comprehensive, multi-faceted RRS re-design aimed to enhance use of NTS at
RRT calls, without the need for dedicated training or additional funding, was implemented.
The re-design drew on themes from the TeamSTEPPS® program [17, 18], and previous
research which described RRS improvement initiatives [14, 19–21]. The present study uses
Interrupted Time Series analysis to investigate the effects of the re-design of an existing RRS.

## Methods

This was a pre-post intervention study assessing the proportion of patients who had repeat
RRT calls before and after implementation of a RRS re-design. Data were collected over a five-

year period prior to the re-design and another five years after its implementation. The present study was part of the *Impact of Non-Technical Skills on Performance and Effectiveness of a Medical Emergency Team (IMPACT)* research program (ClinicalTrials.gov: NCT01551160), components of which have already been reported [11–13].

## Participants

Patients attended by the RRT at a tertiary, outer metropolitan hospital between 1st July 2009 and 30th June 2019 were identified from RRS records. Those who were not admitted to the hospital (e.g. day procedures, outpatients, or visitors) and patients under 18 years of age were ineligible for inclusion.

The cohort of in-patient admissions who were attended by the RRT were divided into two groups: those attended by the RRT more than once during an admission (the 'Repeat Call' group) and those with only one RRT call.

Clinical staff were classified into two groups: those rostered to attend calls as part of the RRT ('members'), and those who recognise clinical deterioration and call the RRT ('users').

## Intervention

The RRS re-design incorporated three components, described in detail previously [13]. These components targeted the key NTS domains of leadership, communication, and co-operation both within the RRT and between RRT members and users.

**1. Regular RRT meetings.** Short meetings for RRT members, designed to address **Leadership** and **Cooperation** within the team, were scheduled to occur at the beginning of each shift. The primary purpose of these meetings was to pre-emptively establish each team member's role and initial task at RRT calls. This approach was designed to avoid spending valuable time doing this at a deteriorating patient's bedside.

**2. Team role badges.** Each member of the RRT was required to wear a badge indicating their role while attending calls. This was designed to reinforce the team **Leadership** role as well as facilitate non-verbal **Communication** of all role designations to RRT members and users present at calls.

**3. RRT members-to-users"hand-off" procedure.** A structured verbal and written process, aiming to improve **Communication** and **Cooperation** between RRT members and users, was introduced for RRT calls ending with the patient remaining on their ward. This formalised the transfer of primary clinical responsibility from the RRT back to the ward team. In particular, the hand-off process encouraged RRT users to voice any ongoing clinical concerns and have them addressed before the RRT departed.

## Study phases

There were two phases of data collection, punctuated by the implementation of the RRS re-design as detailed above. Phase 1 comprised five years (July 2009 –June 2014) and Phase 2 a further five-year period (July 2014 –June 2019). The data presented in this paper were collected retrospectively, extracted at the end of the study from the hospital's RRS and in-patient electronic databases.

Aside from the re-design described above, the configuration and operations of the RRS did not change over the entire study period (i.e. Phase 1 and 2). In particular, the RRT activation criteria, composition of the RRT and provision of Critical Care services at the investigating hospital remained the same throughout.

## Outcome measures

RRT call data, obtained from the hospital RRS database, were aggregated at the per-patient-admission level. Variables were then created to indicate if each admission contained repeat calls, or not, and the count of those repeat calls.

The admission-level data were then collapsed by study month, derived from the date of hospital entry, with month 1 representing July 2009, through to month 120 in June 2019. A variable was created to indicate study phase (Phase 1: months 1–60, Phase 2: months 61–120).

The primary outcome in this study was the proportion of admissions with repeat RRT calls from all admissions with at least one RRT call (per month). This was chosen as an indicator of potentially preventable RRS activity that could be measured throughout both study phases.

The secondary outcome was the mean number of RRT calls per admission (from all admissions with at least one RRT call) to investigate aggregate RRT call load on the hospital.

## Other variables

Demographic data, captured at time of admission, included age, gender, Indigenous identification and socioeconomic status (expressed as a binary variable for Socio-Economic Indexes for Australia (SEIFA) decile of three or less versus greater than three, derived from the 2016 Postal Area Index of Relative Socio-economic Advantage and Disadvantage) [22]. Hospital admission data included elective vs non-elective admission, Charlson Co-Morbidity Index (CCI) and in-patient length of stay (LOS). Counts of hospital admissions during each month of the study were derived from the hospital activity database.

These variables were similarly aggregated by month to account for variations in hospital activity and casemix over the study period. For each study month, the number of admissions, and the percentage of admissions corresponding to male gender, Indigenous identification, SEIFA $\leq$ 3, and non-elective admissions were derived. The mean age, CCI, and hospital LOS were also calculated for each study month.

## Data analysis

Monthly hospital activity and aggregated patient demographics were compared between study phases using Mann-Whitney U-tests.

The effect of the re-design was assessed by Interrupted Time Series (ITS) methodology as described by Bernal et al [23]. In general terms, ITS analyses use segmented regression to compare the observed effect of an intervention, introduced at a defined time point, on an outcome to the effect predicted in the absence of the intervention [24, 25]. ITS quantified the impact of the RRS re-design on the outcomes of interest through the change in coefficients of the fitted regression line at the point of introducing the re-design.

Non-seasonal Auto Regressive Integrated Moving Average (ARIMA) models with a first-order auto-correlation were fit for each outcome variable [26]. Study month was used as the time metric in all models. Initially, simple models were fit that considered only time (study month), phase and the interaction of time and phase (i.e. a different intercept and slope corresponding to the post-intervention phase compared to the pre-intervention phase were allowed for in the regression model–see Fig 2 in Bernal et al [23]). Subsequently, multivariable models that included hospital admission rates, patient demographics and admission characteristics were fit to adjust for any variations between months in hospital activity and casemix over the study phases. The final multivariable model retained variables with a corresponding P-value < 0.1. Sensitivity analyses were also undertaken to examine the impact of potential outliers [27].

Predicted changes in the percentage of repeat call admissions and mean number of calls per admission were derived for each year using the approach outlined in Wagner et al [28]. In this way, the cumulative annual changes in the outcome measures that were attributable to the RRS design were estimated.

Model fit was assessed by the stationary $R^2$ value, where values closer to 1 are indicative of better fit, and the Ljung-Box Q statistic, which indicates if there is a marked lack of fit of the corresponding ARIMA model [29]. Durbin's alternative statistic was used to assess the extent of auto-correlation in the statistical models [30].

Statistical analyses were conducted with SPSS (IBM Corp. Released 2019. IBM SPSS Statistics for Windows, Version 26.0. Armonk, NY: IBM Corp), with the exception of Durbin's alternative statistic, which was calculated using Stata (StataCorp. 2017. Stata Statistical Software: Release 15. College Station, TX: StataCorp LLC).

### Ethics

This study was approved by the Central Adelaide Health Network Human Research Committee (approval number: 2012069).

The need for patient signed consent was waived on the grounds that data used in this study were already collected electronically for hospital quality assurance purposes, no unique patient identifiers were included in the study database, and all individual patient level variables were aggregated by study month prior to analysis and reporting.

## Results

The RRS database provided records for 9754 patients who were attended by the RRT during the study period. From these, 93 paediatric patients and 122 visitors, staff or outpatients were excluded as being ineligible. A further 12 in-patients for whom the database had incomplete records were also excluded. Of the remaining 9527 patient admissions, 3073 occurred in Phase 1 and 6454 in Phase 2. The hospital in-patient database recorded 188016 admissions during Phase 1 and 240910 in Phase 2.

In Phase 2, by comparison to Phase 1, there were more mean hospital admissions per month (4015 [SD 419.7] vs 3134 [SD 222.0], P<0.01) and a greater percentage of those hospital admissions were attended by the RRT (2.6% [standard deviation (SD) 0.5] versus 1.6% [SD 0.4], P<0.01).

Compared to Phase 1, in Phase 2 there were shorter mean in-patient LOS (10.9 days [SD 1.6] vs 12.9 [SD 3.0], P<0.01), lower mean patient age (67.4 [SD 2.0] vs 68.6 [SD 2.8], P<0.01), lower percentage of patients with low socioeconomic status (68.2% [SD 7.0%] vs 79.7% [SD 6.4%], P<0.01) and lower mean CCI (4.5 [SD 0.3] vs 4.8 [SD 0.46], P<0.01). Hospital activity and patient demographic data are summarised by year of the study in Table 1.

### Primary outcome

The ARIMA univariate model estimated the slope as 0.115 (standard error (SE) 0.047) in Phase 1, and 0.029 (SE 0.047) in Phase 2, indicating an observed change in slope between phases of -0.087 (SE 0.067), as shown in Fig 1.

Similar results were found for the final multivariable model, in which proportion of non-elective admissions and average hospital LOS were also retained as covariates. In this model, the change in slope due to the re-design was estimated to be -0.118 (SE 0.067).

The final multivariable model estimated a 6% decrease (95% confidence interval (CI)-15.1–3.1, P = 0.19) in the proportion of RRT attended patients triggering repeat calls (per month) by the fifth-year post-implementation of RRS re-design. The estimated cumulative change in the

**Table 1. Hospital activity and demographic data (for patients having RRT calls) by study year.**

| Study Year | Count of All Admissions | Count of RRT Call Admissions | LOS Mean (SD) | Age Mean (SD) | Male Mean % (SD) | Indigenous Mean % (SD) | Low SEIFA Mean % (SD) | CCI Mean (SD) | Non-Elective Mean % (SD) |
|---|---|---|---|---|---|---|---|---|---|
| 1 | 34238 | 507 | 14.4 (4.2) | 67.8 (3.3) | 48.7 (6.2) | 0.9 (1.3) | 81.7 (6.7) | 4.7 (0.4) | 90.8 (3.1) |
| 2 | 36087 | 506 | 14.0 (3.0) | 69.1 (2.8) | 52.3 (9.4) | 1.8 (2.0) | 79.0 (6.7) | 4.9 (0.5) | 94.4 (2.7) |
| 3 | 37785 | 578 | 11.7 (2.1) | 68.9 (2.6) | 51.4 (7.4) | 3.0 (1.8) | 82.3 (4.5) | 4.8 (0.5) | 93.4 (3.0) |
| 4 | 39441 | 666 | 12.1 (2.2) | 68.0 (3.1) | 50.8 (8.7) | 2.6 (1.7) | 78.4 (8.2) | 4.6 (0.5) | 94.9 (3.2) |
| 5 | 40465 | 816 | 12.6 (2.5) | 69.3 (2.4) | 47.9 (5.2) | 1.7 (1.4) | 77.0 (5.0) | 4.9 (0.4) | 92.9 (2.9) |
| Phase 1 overall | 188016 | 3073 | 12.9 (3.0) | 68.6 (2.8) | 50.2 (7.5) | 2.0 (1.8) | 79.7 (6.4) | 4.8 (0.5) | 93.3 (3.2) |
| 6 | 41098 | 887 | 11.0 (0.6) | 67.2 (2.5) | 52.1 (6.6) | 2.4 (2.1) | 74.5 (5.5) | 4.7 (0.4) | 92.4 (2.7) |
| 7 | 45307 | 1174 | 9.9 (1.5) | 65.8 (2.1) | 45.7 (5.0) | 1.9 (1.2) | 72.4 (6.4) | 4.3 (0.3) | 92.0 (2.8) |
| 8 | 49009 | 1223 | 11.6 (2.0) | 67.8 (1.6) | 49.5 (5.2) | 2.0 (1.6) | 65.5 (5.6) | 4.4 (0.2) | 94.6 (2.6) |
| 9 | 52123 | 1541 | 11.5 (1.8) | 68.2 (1.6) | 49.7 (4.8) | 3.1 (1.7) | 66.6 (4.9) | 4.6 (0.2) | 93.2 (3.3) |
| 10 | 53373 | 1629 | 10.7 (1.3) | 67.9 (1.5) | 49.3 (5.3) | 3.6 (1.8) | 61.8 (4.5) | 4.6 (0.3) | 93.4 (1.9) |
| Phase 2 overall | 240910 | 6454 | 10.9 (1.6) | 67.4 (2.0) | 49.3 (5.6) | 2.6 (1.7) | 68.2 (7.0) | 4.5 (0.3) | 93.1 (2.7) |

RRT = Rapid Response Team, SD = standard deviation, LOS = length of stay, SEIFA = socio-economic indexes for Australia, CCI = Charlson co-morbidity index.

observed percentage of repeat call admissions in Phase 2, compared to the percentage predicted if the re-design had not been implemented, is shown in Fig 2.

Durbin's alternative test statistics were 2.35 on 1 df (P = 0.12) and 2.13 on 1 df (P = 0.14) for the univariate and multivariable models, respectively, and the stationary $R^2$ values were 0.26 for the univariate and 0.30 for the multivariable models, respectively. The Ljung-Box Q statistic indicated there was no significant lack of fit observed for the univariate (15.77 on 17

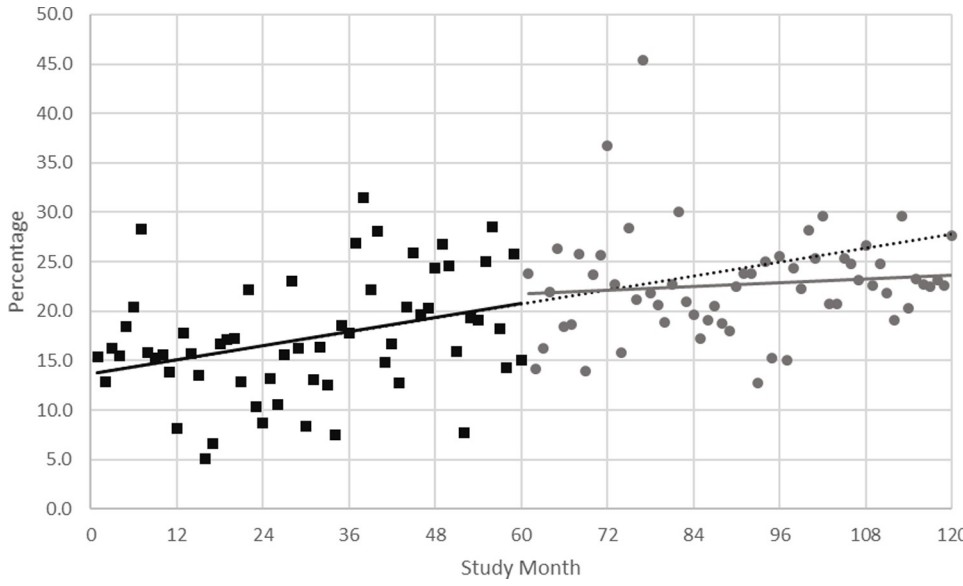

**Fig 1. Percentage of repeat call admissions per month representing the ARIMA univariate model.** Phase 1 monthly observed data in black squares, with slope illustrated by the solid black line. Phase 2 monthly observed data in grey circles, with slope illustrated by the solid grey line. The slope in Phase 1 is extended into Phase 2 and represented by the dotted black line for comparison with Phase 2 observed data.

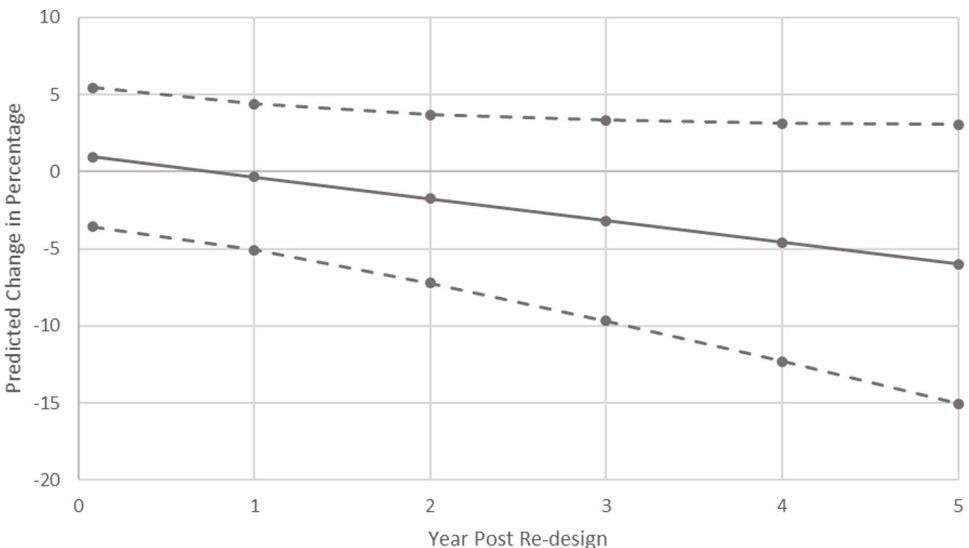

**Fig 2. Final multivariable model.** Cumulative predicted change in percentage of repeat call admissions (per month) associated with the RRS re-design, with 95% confidence intervals.

df; P = 0.54), nor for the multivariable model (Q = 20.83 on 17 df; P = 0.23). Taken together, these statistics suggest reasonable fit of the ITS models.

Given the unusual observation in November 2015 (study month 77), a sensitivity analysis was conducted excluding this value. This analysis resulted in slightly attenuated regression coefficients for the univariate (-0.068 [SE 0.064] vs -0.087 [SE 0.067]) and multivariable models (-0.094 [SE 0.065] vs -0.118 [SE 0.067]), and a modest alteration of the estimated change in percentage of patients having repeat calls versus predicted to -4.9% (95% CI -13.7–3.8, P = 0.27) as shown in S1 Fig.

## Secondary outcome

The change in regression coefficient for the mean number of calls per admission in Phase 2 compared to Phase 1 associated with implementation of the re-design was -0.001 (SE 0.001) in the ARIMA univariate model. Fig 3 shows the observed data for Phase 1 and Phase 2.

At the end of the Phase 2, the final multivariable model, retaining hospital LOS, showed a predicted difference of -0.07 (95%CI -0.23–0.08) calls per admission (P = 0.35) as shown in Fig 4.

The fit statistics from the ITS models for the calls per admission were similar to those observed in the analysis of the primary outcome, again suggesting reasonable fit. Durbin's alternative test statistics were 0.68 on 1 df (P = 0.41) and 0.58 on 1 df (P = 0.45) for the univariate and multivariable models, respectively. The stationary $R^2$ values were 0.26 for the univariate and 0.30 for the multivariable models. The Ljung-Box Q statistic indicated there was no significant lack of fit for the univariate (12.83 on 17 df; P = 0.75), nor for the multivariable model (Q = 15.82 on 17 df; P = 0.54).

A sensitivity analysis excluding the unusual November 2015 observation led to results that were essentially unchanged, with the pre-post regression coefficient change in slope of -0.001 [SE 0.001] and 0.07 fewer predicted calls per admission (95%CI -0.21–0.07, P = 0.34), as presented in the final multivariable model in S2 Fig.

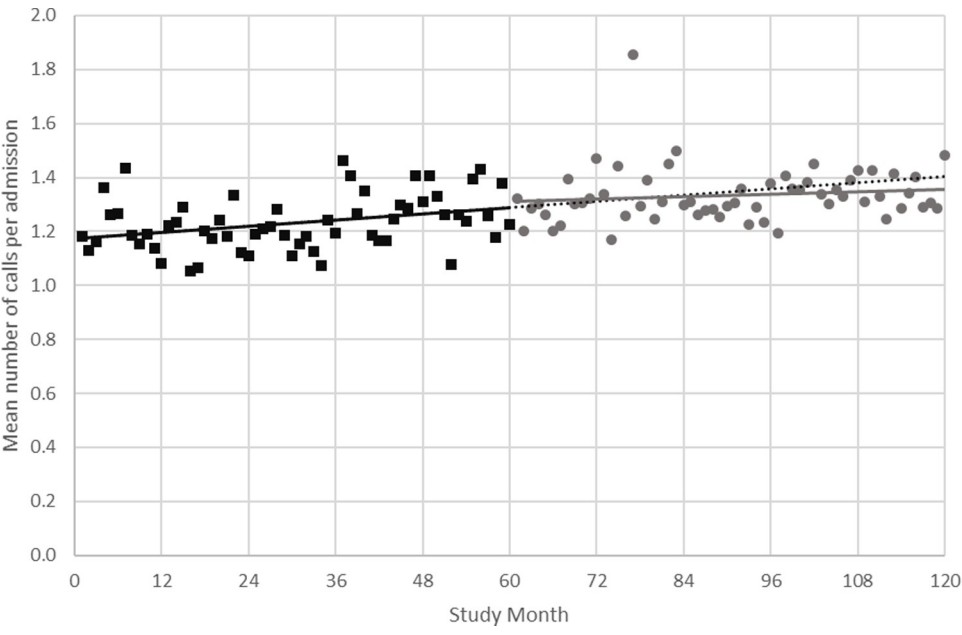

**Fig 3. Mean number of calls per admission by study month for the ARIMA univariate model.** Phase 1 observed data in black squares, with slope illustrated by the solid black line. Phase 2 observed data in grey circles, with slope illustrated by the solid grey line. The Phase 1 slope is extended into Phase 2 and represented by the dotted black line for comparison with Phase 2 observed data.

## Discussion

### Key findings

Following a multi-faceted RRS re-design, modest, but not statistically significant, reductions were estimated in the percentage of RRT-attended patients having repeat calls and the average number of repeat RRT calls per patient, with changes in hospital activity and patient demographics accounted for in the statistical analyses. The observed reduction saw six percent fewer

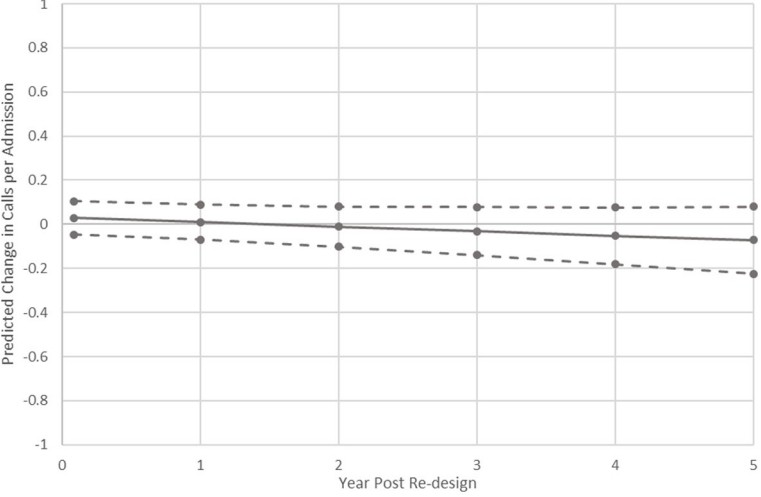

**Fig 4. Final multivariable model.** Cumulative predicted change in calls per admission associated with the RRS re-design, with 95% confidence intervals.

RRT attended patients going on to have repeat calls (per month). In the context of a median 30-minute call duration [12], this would be equivalent to a reduction in RRS activity of three hours per 100 patients attended by the RRT.

## Interpretation of results

Notably, the investigating hospital saw markedly increased activity throughout the study period. This partly explains the disparity in the number of subjects between the two study Phases. However, there was also a statistically significant increase in the percentage of admissions subject to RRT calls in Phase 2 versus Phase 1. Reviews of RRS operational activity observations have reported that increases in RRT calling over time following introduction of an RRS are commonplace [1–7]. A likely cause is familiarity and acceptability that reduce barriers to calling as the RRS matures.

In this study, the changes in hospital activity and patient demographics were accounted for in the multivariable Interrupted Time Series analyses. The estimated reduction in proportion of patients experiencing a repeat call following the RRS re-design has potential implications for patient mortality. A previous publication from this research program and two other studies corroborated the association between repeat calling and mortality [12, 31, 32].

## Implications of results

For organisations, there are two important potential benefits to operational efficiency from exploring potentially avoidable repeat calling. First, RRTs tend to draw resources from other acute clinical unit rosters, such as ICU and Internal Medicine, rather than have their own supernumerary staffing [8, 9]. Therefore, even modest reductions in potentially avoidable re-calling of the RRT allow staff more time to attend to their primary rostered clinical duties.

Second, as RRS activity increases, there is a proportionate potential for concurrent RRT calls. Most RRS only roster a single RRT [8, 9], which presents a risk to simultaneously deteriorating patients from delayed or incomplete attendance by an RRT. Thus, attempts to reduce the likelihood of avoidable repeat calls may help to ensure constant RRS capability to attend clinical deterioration promptly and effectively.

## Contribution to evidence base

The literature on re-designing the RRS to improve use of NTS during RRT calls is scant. Most published articles reinforce simulation training as the gold standard mechanism to achieve this [14, 15]. Staff training is labour and cost intensive, so alternative strategies need to be explored.

Kansal et al. evaluated streamlining information sharing by ward staff to the RRT on their arrival to calls, alongside other restructuring of their respective RRS [19]. Although they did report reductions in rates of unexpected deaths and other adverse patient outcomes after re-designing the RRS, these authors could not ascribe the role of the enhanced handover as the sole reason for these improvements due to changes to a tiered RRS response taking place at the same time.

Prince et al. and Mardegan et al. described changes to operations of the RRT during calls [20, 21]. Prince et al. focused on visual identification of team member roles during cardiac arrest calls which was incorporated into simulation training for the RRT. These authors noted perceived improvements in communication during RRT calls, although no pre-training data were collected. This reflects improvements in perceptions and experiences of interactions during RRT calls we found in a previous publication from this research program [13]. Mardegan et al. only described staff satisfaction after introduction of a RRT call checklist that facilitated

handover from ward staff to the RRT on their arrival to calls. While staff were positive in general about each of these interventions, no effects on patient outcomes were presented in either study [20, 21].

The present study reports implementation of a multi-faceted RRS re-design that aimed to promote use of NTS during RRT calls. While not statistically significant, the results may still be worthy of consideration at an organisational level, especially given the negligible barriers or overheads to implementing the three components of this RRS re-design.

## Strengths and limitations

To the best of our knowledge, this is the first description of objective RRS performance outcomes measured around implementation of a Non-Technical Skills focused system re-design.

The Interrupted Time Series approach is particularly helpful for studying organisation-level interventions where randomised controlled trials are infeasible [23–25]. Its use in this study allowed us to investigate the effect of the re-design on the outcomes and accounted for temporal trends and variations [28]. The analysis also demonstrated that the effect of the re-design in reducing rates of repeat calling was sustained throughout Phase 2, with no evidence of attrition of benefit.

As with any pragmatic study, there are limitations. First, we acknowledge the absence of results regarding RRS compliance with the components of the re-design, or usage of NTS during RRT calls. Due to limited financial resources for the study, it was not possible to employ observers to objectively record attendance at RRT meetings, wearing of badges or usage of NTS during RRT calls, and adherence to the required hand-off process at RRT call completions.

Second, although a range of demographic and hospital activity co-variates were included in the analyses and the configuration of RRS did not otherwise alter during the entire study period, it is still possible that some other unmeasured factors, such as seniority of RRT clinicians, could have influenced the findings.

Finally, some repeat calls may indicate a correctly functioning RRS responding to clinically discrete deteriorations. However, this study focused on the wider resourcing implication for organisations, and so did not separate these from the preventable calls. All repeat calls present a potential logistical and staffing burden on hospitals, so that even modest improvements, such as observed here, may confer benefits to the organisation.

## Future scope for re-designing the RRS

The RRS re-design used in this study was developed with the understanding that further iteration and re-evaluation would be worthwhile. Some potential revisions to the re-design, such as role stickers, rather than badges, and electronic availability of RRT rosters, have already been proposed in a previous publication from this research program [13].

Further to those, a natural addition to the RRS would be debriefs for the RRT and other hospital staff involved in calls [33, 34]. This could take one of two forms: "hot debrief" conducted immediately after completion of each RRT call or "cold debrief" in which cases are reviewed later at scheduled meetings [34]. There are challenges in implementing either of these debrief methods. Hot debrief depends on RRT members, and possibly also ward staff, involved in that call remaining available to attend. For ad-hoc RRTs rostered from other clinical roles, this may be infeasible [8, 9]. The scheduled, delayed nature of cold debrief provides more opportunity for RRT members to plan their attendance and avoid conflicts with other clinical duties, so may be easier to implement, but all RRT members are unlikely to be rostered to work at the scheduled time of the cold debrief [34].

### Context within the IMPACT research project

As outlined earlier, this study was conducted as part of a larger research project. In a parallel survey study of perceptions and experiences of NTS use during RRT calls of RRT members and those calling the RRT (users), this RRS re-design was associated with significant reductions in reported experience of conflict [13]. Furthermore, both in quantitative data and free-text comments, improvements in leadership, communication and cooperation between RRT members and users during RRT calls were reported following introduction of the re-design.

Thus, the apparent lack of effect of the RRS re-design on the proportion of admissions with repeat RRT calls and the mean number of RRT calls per admission raises the question of whether organisational change did not occur, or whether a potential improvement (as suggested by the survey findings) was not captured by the outcomes used here. Therefore, as part of future research, identification and use of other outcome measures that are more sensitive to NTS performance during RRT calls should be explored.

### Conclusions

This study reports a multi-faceted RRS re-design which was associated with a modest, but not statistically significant, reduction in the percentage of patients per month having repeat calls and the average number of repeat calls per admission.

In an era of economic and health workforce constraints, even small potential improvements may still have relevance to organisations. This RRS re-design (and assessment thereof) has scope for further refinement, and may be of interest to RRS clinicians and managers seeking to implement their own pragmatic, low-cost quality improvement initiatives.

### Supporting information

**S1 Fig. Percentage of repeat call admissions per month representing the ARIMA univariate model with outlier detection enabled.** Phase 1 observed data shown as black squares, with trend shown as the solid black line. Phase 2 observed data shown as grey circles, with trend shown as the solid grey line. The Phase 1 trend is extended into Phase 2 as the dotted black line for comparison with Phase 2 observed data. Observation for study month 77 (November 2015) was identified as an outlier and excluded for this sensitivity analysis.
(TIF)

**S2 Fig. Mean number of calls per admission by study month for the ARIMA univariate model with outlier detection enabled.** Phase 1 observed data shown as black squares, with trend shown as the solid black line. Phase 2 observed data shown as in grey circles, with trend shown as the solid grey line. The Phase 1 trend is extended into Phase 2 as the dotted black line for comparison with Phase 2 observed data. Observation for study month 77 (November 2015) was identified as outlier and excluded for this sensitivity analysis.
(TIF)

**S1 Data. De-identified, by-month aggregated data used in statistical analyses.**
(CSV)

### Acknowledgments

Dr Bill Wilson, Chief Medical Information Officer, Northern Adelaide Local Health Network, Adelaide, Australia for assistance with data extraction from hospital electronic databases.

## Author Contributions

**Conceptualization:** Richard Chalwin.

**Data curation:** Richard Chalwin, Victoria Eaton.

**Formal analysis:** Richard Chalwin, Lynne Giles.

**Investigation:** Richard Chalwin, Victoria Eaton, Lynne Giles.

**Methodology:** Richard Chalwin, Amy Salter, Jonathan Karnon, Lynne Giles.

**Project administration:** Richard Chalwin, Victoria Eaton.

**Resources:** Victoria Eaton.

**Supervision:** Amy Salter, Jonathan Karnon, Lynne Giles.

**Validation:** Amy Salter, Jonathan Karnon, Lynne Giles.

**Writing – original draft:** Richard Chalwin.

**Writing – review & editing:** Richard Chalwin, Amy Salter, Jonathan Karnon, Victoria Eaton, Lynne Giles.

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
