## [Decision Letter · Decision Letter 0]

14 Oct 2021

PONE-D-21-18171

Effect of a multi-faceted Rapid Response System re-design on repeat calling of the Rapid Response Team

PLOS ONE

Dear Dr. Chalwin,

Thank you for submitting your manuscript to PLOS ONE. After careful consideration, we feel that it has merit but does not fully meet PLOS ONE’s publication criteria as it currently stands. Therefore, we invite you to submit a revised version of the manuscript that addresses the points raised during the review process.

We look forward to receiving your revised manuscript.

Kind regards,

Tai-Heng Chen, M.D.

Academic Editor

PLOS ONE

3.We note that you have indicated that data from this study are available upon request. PLOS only allows data to be available upon request if there are legal or ethical restrictions on sharing data publicly. For more information on unacceptable data access restrictions, please see http://journals.plos.org/plosone/s/data-availability#loc-unacceptable-data-access-restrictions.

Reviewers' comments:

Reviewer's Responses to Questions

**Comments to the Author**

1. Is the manuscript technically sound, and do the data support the conclusions?

Reviewer #1: Partly

Reviewer #2: Partly

Reviewer #3: Yes

2. Has the statistical analysis been performed appropriately and rigorously? 

Reviewer #1: Yes

Reviewer #2: Yes

Reviewer #3: Yes

3. Have the authors made all data underlying the findings in their manuscript fully available?

Reviewer #1: Yes

Reviewer #2: Yes

Reviewer #3: Yes

4. Is the manuscript presented in an intelligible fashion and written in standard English?

Reviewer #1: Yes

Reviewer #2: Yes

Reviewer #3: Yes

5. Review Comments to the Author

Reviewer #1: 1. Cite the source (reference) when models proposed by other researchers are mentioned.

2. Conclusion- Need major revision. Should include the key findings, answer the research questions, highlight the limitations, and future scope, etc.

3. Interpretation of Results: Use the reference numbers instead of author names.

Reviewer #2: The manuscript is technically sound and the statistical analyses performed are appropriate. Just one question regarding the conclusion, I could not understand how your findings are 'suggestive of feasibility' of a redesign. Please kindly elaborate on that.

Please edit the article for grammar. There are numerous minor grammatical errors; there is no easy way to list them all here.

As examples: In the abstract:

Background: ...training; however, this...

Methods:... effect of AN RRS re-design.

Reviewer #3: Dear Author(s)

The design of the study is original, well-defined problems, and clearly reveals the purpose or hypothesis. Methods and practices are clear and understandable. The findings are discussed appropriately and adequately. The language of writing is clear and grammatically well written. My suggestion is to accept as it.

Best Regards

6. PLOS authors have the option to publish the peer review history of their article (what does this mean?). If published, this will include your full peer review and any attached files.

Reviewer #1: No

Reviewer #2: No

Reviewer #3: No

---

## [Author Response · Author response to Decision Letter 0]

2 Nov 2021

Reviewer 1 Comments

1. “Cite the source (reference) when models proposed by other researchers are mentioned”

A reference is provided for ARIMA statistical modelling. The Cochrane (EPOC) reference has been updated to an original description of the segmented regression Interrupted Time Series analytical technique used.

2. “Conclusion- Need major revision”

The conclusion has been rewritten per your request to include the key findings and scope for future research. It now reads:

This study reports a multi-faceted RRS re-design which did not demonstrate a statistically significant effect on the percentage of patients per month having repeat calls or the number of repeat calls per admission.

The RRS re-design presented herein may be of interest to RRS clinicians and managers as a pragmatic, low-cost quality improvement initiative that could inspire future efforts to improve operational efficiency of the RRS.

3. “Interpretation of Results: Use the reference numbers instead of author names”

The names of the lead authors of the two cited papers have been removed.

Reviewer 2 Comments

1. “I could not understand how your findings are 'suggestive of feasibility' of a redesign”

That statement in Conclusions has been removed. The comment regarding the re-design has been rewritten to better reflect the potential contribution of the study to future research.

The conclusion now reads:

This study reports a multi-faceted RRS re-design which did not demonstrate a statistically significant effect on the percentage of patients per month having repeat calls or the number of repeat calls per admission.

The RRS re-design presented herein may be of interest to RRS clinicians and managers as a pragmatic, low-cost quality improvement initiative that could inspire future efforts to improve operational efficiency of the RRS.

2. “Please edit the article for grammar”

We have thoroughly perused the manuscript and used Microsoft Word grammar checks. Errors have been corrected per PLOS One style and language requirements.

Reviewer 3 Comments

None to address

---

## [Decision Letter · Decision Letter 1]

19 Dec 2021

PONE-D-21-18171R1Effect of a multi-faceted Rapid Response System re-design on repeat calling of the Rapid Response TeamPLOS ONE

Dear Dr. Chalwin,

Thank you for submitting your manuscript to PLOS ONE. After careful consideration, we feel that it has merit but does not fully meet PLOS ONE’s publication criteria as it currently stands. Therefore, we invite you to submit a revised version of the manuscript that addresses the points raised during the review process. Please submit your revised manuscript by Feb 02 2022 11:59PM. If you will need more time than this to complete your revisions, please reply to this message or contact the journal office at plosone@plos.org. Please include the following items when submitting your revised manuscript:A rebuttal letter that responds to each point raised by the academic editor and reviewer(s). You should upload this letter as a separate file labeled 'Response to Reviewers'.A marked-up copy of your manuscript that highlights changes made to the original version. You should upload this as a separate file labeled 'Revised Manuscript with Track Changes'.An unmarked version of your revised paper without tracked changes. You should upload this as a separate file labeled 'Manuscript'.If applicable, we recommend that you deposit your laboratory protocols in protocols.io to enhance the reproducibility of your results. Protocols.io assigns your protocol its own identifier (DOI) so that it can be cited independently in the future. For instructions see: https://journals.plos.org/plosone/s/submission-guidelines#loc-laboratory-protocols. Additionally, PLOS ONE offers an option for publishing peer-reviewed Lab Protocol articles, which describe protocols hosted on protocols.io. Read more information on sharing protocols at https://plos.org/protocols?utm_medium=editorial-email&utm_source=authorletters&utm_campaign=protocols.

We look forward to receiving your revised manuscript.

Kind regards,

Tai-Heng Chen, M.D.

Academic Editor

PLOS ONE

Journal Requirements:

Reviewers' comments:

Reviewer's Responses to Questions

**Comments to the Author**

1. If the authors have adequately addressed your comments raised in a previous round of review and you feel that this manuscript is now acceptable for publication, you may indicate that here to bypass the “Comments to the Author” section, enter your conflict of interest statement in the “Confidential to Editor” section, and submit your "Accept" recommendation.

Reviewer #1: All comments have been addressed

Reviewer #2: All comments have been addressed

2. Is the manuscript technically sound, and do the data support the conclusions?

Reviewer #1: Yes

Reviewer #2: Yes

3. Has the statistical analysis been performed appropriately and rigorously? 

Reviewer #1: Yes

Reviewer #2: Yes

4. Have the authors made all data underlying the findings in their manuscript fully available?

Reviewer #1: Yes

Reviewer #2: Yes

5. Is the manuscript presented in an intelligible fashion and written in standard English?

Reviewer #1: Yes

Reviewer #2: Yes

6. Review Comments to the Author

Reviewer #1: Conclusion: Need revision.

A conclusion should address: answers research questions/objectives; explains discrepancies and unexpected findings; states importance of discoveries and future implications.

Reviewer #2: The study is methodologically sound and the authors have revised the manuscript to adequately address the concerns I had.

7. PLOS authors have the option to publish the peer review history of their article (what does this mean?). If published, this will include your full peer review and any attached files.

Reviewer #1: **Yes: **R. S. Ajin

Reviewer #2: No

---

## [Author Response · Author response to Decision Letter 1]

20 Jan 2022

Reviewer 1 Comments

1. “Conclusion: Need revision. A conclusion should address: answers research questions/objectives; explains discrepancies and unexpected findings; states importance of discoveries and future implications.”

The conclusions of both the abstract and manuscript have been thoroughly revised and rewritten per your request to clarify the study objectives, key findings, discrepancies and the scope for iterative research and quality improvement.

The abstract conclusion now reads:

“This study of an RRS re-design showed modest, but not statistically significant, reductions in the proportion of admissions with repeat calls and the mean number of repeat calls per admission. Given the economic and workforce capacity issues that all health care systems now face, even small improvements in the RRS may have lasting impact across the organisation. For the potential interest of RRS managers, this paper presents a pragmatic, low-cost initiative intended to enhance communication and cooperation at RRT calls.”

The manuscript conclusion now reads:

“This study reports a multi-faceted RRS re-design which was associated with a modest, but not statistically significant, reduction in the percentage of patients per month having repeat calls and the average number of repeat calls per admission.

In an era of economic and health workforce constraints, even small potential improvements may still have relevance to organisations. This RRS re-design (and assessment thereof) has scope for further refinement, and may be of interest to RRS clinicians and managers seeking to implement their own pragmatic, low-cost quality improvement initiatives.”

Reviewer 2 Comments

None to address

---

## [Decision Letter · Decision Letter 2]

3 Mar 2022

Effect of a multi-faceted Rapid Response System re-design on repeat calling of the Rapid Response Team

PONE-D-21-18171R2

Dear Dr. Chalwin,

We’re pleased to inform you that your manuscript has been judged scientifically suitable for publication and will be formally accepted for publication once it meets all outstanding technical requirements.

Kind regards,

Tai-Heng Chen, M.D.

Academic Editor

PLOS ONE

Reviewers' comments:

Reviewer's Responses to Questions

**Comments to the Author**

1. If the authors have adequately addressed your comments raised in a previous round of review and you feel that this manuscript is now acceptable for publication, you may indicate that here to bypass the “Comments to the Author” section, enter your conflict of interest statement in the “Confidential to Editor” section, and submit your "Accept" recommendation.

Reviewer #1: All comments have been addressed

2. Is the manuscript technically sound, and do the data support the conclusions?

Reviewer #1: Yes

3. Has the statistical analysis been performed appropriately and rigorously? 

Reviewer #1: Yes

4. Have the authors made all data underlying the findings in their manuscript fully available?

Reviewer #1: Yes

5. Is the manuscript presented in an intelligible fashion and written in standard English?

Reviewer #1: Yes

6. Review Comments to the Author

Reviewer #1: All the comments have been properly addressed by the authors. My decision is to accept the manuscript.

7. PLOS authors have the option to publish the peer review history of their article (what does this mean?). If published, this will include your full peer review and any attached files.

Reviewer #1: No

---

## [Editor Report · Acceptance letter]

8 Mar 2022

PONE-D-21-18171R2 

Effect of a multi-faceted Rapid Response System re-design on repeat calling of the Rapid Response Team 

Dear Dr. Chalwin:

I'm pleased to inform you that your manuscript has been deemed suitable for publication in PLOS ONE. Congratulations! Your manuscript is now with our production department. 

Kind regards, 

on behalf of

Dr. Tai-Heng Chen 

Academic Editor

PLOS ONE